# Social Support and Dietary Habits as Anxiety Level Predictors of Students during the COVID-19 Pandemic

**DOI:** 10.3390/ijerph18168785

**Published:** 2021-08-20

**Authors:** Dorota Ortenburger, Dariusz Mosler, Iuliia Pavlova, Jacek Wąsik

**Affiliations:** 1Department of Kinesiology and Health Prevention, Jan Dlugosz University of Czestochowa, 42200 Czestochowa, Poland; d.ortenburger@gmail.com (D.O.); dariusz.mosler@gmail.com (D.M.); 2Theory and Methods of Physical Culture Department, Lviv State University of Physical Culture, 79007 Lviv, Ukraine; pavlova.j.o@gmail.com

**Keywords:** STAI, anxiety, social support, nutrition habits

## Abstract

The COVID-19 pandemic is a public health emergency concern and a challenge to students’ mental health due to changes in education and social isolation. The aim of this research was to expand knowledge about the relations that shape the level of anxiety amongst men and women who are studying during the pandemic in terms of the relations towards their sense of social support and their nutritional behaviors. A State–Trait Anxiety Inventory was used to measure anxiety level, alongside supplementary questions such as the feeling of support from close ones, concentration of attention on nutrition during the pandemic and externally derived factors (university, specialization). Analysis of the regression was applied to the examination of the dependency between the anxiety level (in both forms of its occurrence—as state-anxiety and as trait-anxiety). We observed that the pandemic situation affected a level of state-anxiety above average (mean value of 46–48 points) even when students felt social support. Nutrition habits and chosen education type are associated with trait-anxiety level, which was also elevated (mean values of 49–50 points). Chosen factors had a partial influence on the anxiety level of students, therefore their mental health should concern shaping positive nutrition habits and social support.

## 1. Introduction

There are numerous traits which affect human ability to handle difficult and stressful situations [1]. In 2020, almost all countries were affected by the COVID-19 pandemic. In order to stop its spread, governments decided to lockdown and isolate people [2]. Although it seemed like the only proper way of conduct in such a situation, social distancing and changes in daily life activity can affect personal wellbeing, mental health and results in an increase in depression and anxiety symptoms [3,4]. SARS-CoV-2 virus may cause severe somatic symptoms affecting mainly pulmonary functions, but also the nervous system. In some cases, infection is deadly. This threat not only has an impact on somatic, but also on mental health. The pandemic situation is associated with increased fear and anxiety, therefore among various groups of people there has been an increase in mental health problems. Social isolation affects students, who—alongside other groups—find escape in substance abusive behaviors [5]. Psychiatrists and psychologists began researching the severity of mental health problems and started to develop proper interventions to handle these issues, such as shifting towards online and telecare programs [6]. Researchers from across the world are making efforts to collect and publish data about mental health status from different groups alongside selected behaviors associated with pandemics to help psychiatrists gain better knowledge and help them develop proper programs, not only to take care of people with mental disorders, but also as prophylaxis interventions to limit the development of such disorders for people who are more vulnerable to them [7]. Pandemic and social isolation results in worse dietary habits [8] and lower physical activity [9,10]. It is crucial to maintain proper eating habits and physical activity, as well as positive social relations, to maintain a high quality of life [11].

Among the affected groups, students experienced substantial changes in their curriculum with changes to online versions of studying in social isolation. Sudden changes in conditions affected students’ mental health [12]. Their reaction to the pandemic was expressed individually with a complex spectrum of emotions and behaviors. Humans can maintain control of some behaviors while controlling the strength of individual reactions to long-term situations, which could be observed before the pandemic in 2020 [13]. One of the aspects that individuals can control to a certain degree is eating behaviors and dietary habits with a product choice [14] and regulating its amount [15].

The phenomenon of eating disorders during the COVID-19 pandemic and an increase in anxiety levels among many social groups [16], especially students, brings our attention to the relation between these factors. We hypothesize that lower anxiety levels should be corresponding to better dietary habits and resistance to poor eating habits during feelings such as anxiousness or distress. Social support should also play a significant role during stress management. Anxiety levels of different groups of students during the pandemic were tested [17], but their relation to mentioned habits and environmental conditions are not fully explored. This global pandemic also brings up relatively new terms regarding mental health. Health anxiety refers to fear of catching or having a disease, as infection can cause no visible symptoms, but a person who has it can pass it to other people. Therefore, anxiety could be connected with changes in everyday life activities and also with direct fear of sickness of oneself and others. In some doses, this fear is normal and allows us to behave rationally during this crisis, but in some cases it can turn into a compulsive disorder or total negligence of threat, which can represent both extremes of damaging behavior regarding mental health [18].

Individual proneness to stress, somatic and psychological responses of a person could be perceived as a trait [15,19]. Individual characteristics are connected to stress management, which is a way of behaving while facing a sudden stressful situation or during prolonged stress exposition [16,20]. Traits are more constant features of a person, such as their state of somatic health, skills, behaviors and psychosomatic stress response while feeling hopeless. On the other hand, there are outer factors of stress management referring to the general social environment of the individual. This concerns social relations, support from close people and loved ones or financial status [21]. The role of social support during extreme conditions such as a pandemic seems to be crucial in terms of stress management [22].

In the scope of mentioned premises, we state that the aim of this study is to verify how biological variables, such as declared gender or university students attend, plays a significant role in the anxiety level and if this division makes diverse profiles of attitudes towards eating behaviors and sense of social support during the COVID-19 pandemic.

## 2. Materials and Methods

### 2.1. Participants

The study includes 1012 participants recruited from 3 different universities: National Academy of the National Guard of Ukraine, Lviv State University of Physical Culture and Jan Dlugosz University in Czestochowa. The first two are located in Ukraine, while the third one is located in Poland. Tested group was composed of 535 women and 477 men. Participants were students from 18 to 24 years old, from different specializations. Data were collected during winter academic semester of 2020/21 (from November to end of February). During this period there were ongoing lockdowns with restrictions of online or hybrid learning modes. In Poland, there was a second wave of pandemic with 7-day moving average of 25,500 active cases at peak till around 7000 at its end in February. In the Ukraine, peak of the wave was at the end of November with 7-day moving average of 13,900 active cases and around 5000 daily cases at the end of February.

### 2.2. Measurement Tools

Anxiety levels were measured by State–Trait Anxiety Inventory (STAI) designed for testing adolescents and adults [23]. We used a validated language version of the original inventory to polish language [24]. This inventory measures anxiety divided into two aspects. One aspect is perceived as state-anxiety, which refers to current and ongoing feelings and anxiousness of a person. The second is trait-anxiety, which refers to more constant predispositions of a person, covering internal biopsychosocial components of stress management. Each item was composed of 20 questions with responses scored from 1 to 4 points. Therefore, point range was from 20 to 80 for each subtest. There were set cut points for both state-anxiety (s-anxiety) and trait-anxiety (t-anxiety). S-anxiety scale average cut point was set on 36–38 points in the original handbook [25]. T-anxiety scale average cut point was set to 40–43 points, which corresponds with later studies also setting the cut point on a level of 40–41 points [26,27].

In order to obtain information about sense of social support and diet, additional questions were put beside STAI questionnaire. Three questions were about sense of support from a close person, attitude towards positive dietary habits understood as maintaining healthy eating behaviors and susceptibility towards bad diet habits understood as unhealthy eating behavior during distress and anxiousness. Scores were put in the Likert scale from 1 to 4 points, where 1 was the lowest and 4 was the highest score. First of those two questions regarding attitude towards eating was positive, which means the higher score, the better the person felt about their attitude toward maintaining proper eating habits in the time of pandemic. The second was about negative attitude, therefore the higher they scored, the more prone they were to succumb to unhealthy eating habits during pandemic. Those supplementary questions served as predictors and explanative factors for anxiety levels as main independent variable used in this study. Such questions were commonly practiced in the other anxiety-related studies of students during pandemic [28,29].

### 2.3. Statistical Analysis

Standard descriptive statistic variables were computed, such as mean, minimum and maximum values and standard deviation. Regression models for dependent variables of state-anxiety and trait-anxiety were used. Models were shaped by predictors composed of students’ answers to supplementary questions. Models were validated by comparison with results of Kruskal–Wallis ANOVA analysis by categorial variables of gender and university type. Pairwise comparison was performed as a post-hoc test for three-group comparison. Three-dimensional categorized surface graphs were created to verify influence intensity on predictive variables for independent anxiety variables in regression models. Categorization included both s-anxiety and t-anxiety results with borders set on the average score of both items (36 and 41 points, respectively, as assumed average value), which created variable intensity distribution in four possible sets. All computation was performed using Statistica 13.3 (TIBCO Software Inc., Palo Alto, CA, USA).

## 3. Results

### 3.1. Description of Raw Results

The total sample counted 1012 students, with quite an equal split among genders; 52.8% of students were females. The largest number of students were from Jan Dlugosz University in Czestochowa in Poland (*N* = 377), the second group was from Lviv State University of Physical Culture (*N* = 341) and the smallest number from the National Academy of the National Guard (*N* = 294). In terms of specialization, the largest groups were from physiotherapy (*N* = 379) and physical education (*N* = 245), while the fewest were from command and staff (*N* = 14), law (*N* = 68) and military administration (70). Detailed data are presented in the Table 1.

Female students obtained a broader range of scores, covering almost the whole scale for state-anxiety from 21 to 78 points. Trait-anxiety of females was also in a broad range but the highest value was lesser, with 69 points at maximum. Male students expressed a more narrow range of scores for both state-anxiety (20 to 63 points) and trait-anxiety (33 to 66 points). Despite visible differences in the score dispersity between genders, the mean values of both STAI items were similar for both groups (range from 46.96 points to 49.59 points out of 80) (Figure 1).

With group division by university, the lowest dispersion of scores for both state- and trait-anxiety was revealed by students of the National Academy of the National Guard of Ukraine (s-anxiety, 20 to 60 points and t-anxiety, 35 to 66 points). The highest dispersion of scores were observed for students of Jan Dlugosz University in Czestochowa (s-anxiety: 21 to 78 points and t-anxiety: 20 to 69 points). Division by university showed higher differences in the mean scores of both items, where the lowest mean was for students of Jan Dlugosz University in Czestochowa (45.64 to 46.47 points), while the highest was for students of Lviv State University of Physical Culture (49.13 to 49.72 points).

With division by specialization, the highest range of scores were obtained by students from physical education and physiotherapy (from 20 to 78 points in state-anxiety). A significantly lower range of scores were obtained by students from specializations such as military administration (27 to 66), command and staff (42 to 57), coach (37 to 63) or state security (35 to 64 points). Despite such differences in the range of scores, all groups held a similar mean score level to their respective division for genders and university in a range from 45.56 to 50.26 points out of a maximum of 80.

Scores of students from all supplementary questions ranged from 1 to 4 points. For both males and females, the mean score of sense of support of a close person was the highest (mean from 3.21 to 3.51), while dietary habits had the lowest mean scores (from 1.87 to 1.91 points) (Figure 2). Students of the National Guard of Ukraine had the highest mean sense of support from a close person (3.66 points), while students of the Lviv State University of Physical Culture (3.20 points). In terms of nutrition habits, the lowest scores of dietary habits and highest scores for poor diet in distress, namely, the worst scores were obtained by students from Jan Dlugosz University in Czestochowa. The best habits, expressed by highest mean scores, were showed by students of Lviv State University of Physical Culture. In terms of division by specialization, the lowest sense of support from close persons was expressed by students from physiotherapy and physical education and coach (mean value around 3.2 to 3.35 point), while students of law, state security, military administration and command and staff exceeded a mean value of 3.5 points.

### 3.2. Regression Models Based on Biological and Social Categorial Variables

Regression models were grouped by gender and university separately. For a group of female students, the only statistically significant predictor was self-reported poor diet while in distress. Regression models of male participants had two statistically significant predictors: support of a close person and poor diet while in distress for state-anxiety inventory and dietary habits and poor diet while in distress for trait-anxiety. In the trait-anxiety model for female students, the regression model had worse fit (R = 0.13), while in other cases models had R = 0.20 (Table 2).

The entire structure of predictors analyzed by us clarifies a small part of the variance of the variability of the level of anxiety (despite the confirmed statistical significance). In the acquired explanatory power of model 1 and model 2 (grouped by gender of participants in Table 2) there was no high level of explanatory power, thus the interpretation of the dependency must be cautious. However, the acquired findings expand knowledge to a certain extent on the subject matter of the dependency between the level of anxiety and the approach to nutrition in times of stress, and the feeling of support from closed ones. In the interpretation of this finding, it is necessary to emphasize that in the presented data of this work none of the predictors formulated in the equation of regressions operated independently. The impact of the particular predictors may be hampered by other factors in this model.

Every model divided by university was different and all predictors were significant at least once, except for the state-anxiety model for Lviv State University of Physical education, where none of the predictors were statistically significant, with the lowest fit (R = 0.14). The state-anxiety model of Jan Dlugosz University in Czestochowa had two significant predictors (support of close person and poor diet while in distress) and the best fit among all presented models with R = 0.34. Every analyzed university had at least one better fit model than division by gender (Table 3). Although the results that the regression model of state-anxiety for Lviv State University of Physical Culture did not reach the required significance level, all of the predictors were close to being significant. This was also the case in the National Academy of the National Guard of Ukraine for the trait-anxiety model, where the *p*-value was 0.051.

### 3.3. Model Validation by Kruskal–Wallis ANOVA Analysis Results

Differences in students’ anxiety levels for both STAI items were significantly different with division by gender and university. For the supplementary questions, only student results of question regarding support of a close person were significantly different for both gender and university. Students’ results of the remaining questions were significantly different only by university (Table 4).

Post-hoc test pairwise comparison results of Kruskal–Wallis ANOVA analysis by university showed that most variables differed significantly between all groups. Only results of s-anxiety variable did not differ between universities from Ukraine (Table 5).

Composition of predictive variables (supplementary questions results) intensity were put into four variants, based on both STAI items results below and above mean standardized results. Each graph presents a distinguishable pattern in those configurations: both state- and trait-anxiety below average, state-anxiety below average and trait-anxiety above, trait-anxiety below average and state-anxiety above, both state- and trait-anxiety above average. The highest dispersion of predictors values is visible in the variant of state-anxiety below and trait-anxiety above average ranging from below 2 points to values above 6 points. Surface of the variant with both STAI items above average is the most flattened alongside lowest scores between 2 and 4 points (Figure 3).

## 4. Discussion

Female student results of both STAI items spread almost through all the scale of point (from 1st to 100th percentile), indicating distribution close to normal. Male results were more focused but also with a peak in the middle of distribution. Mean values for each anxiety type were around 47–50 points, which exceeds the average value for this age by almost 10 points [23], which were significantly different in one-sample *t*-test (*p* < 0.00). Students mean values were placed in the 88th percentile in the norms from more than 40 years ago. Considering that more than half of the students were attending courses related to heath science such as physical education, coach or physiotherapy, those results could be compared to students of medical studies. Results of students from this study are similar to those of German medical students, with a mean of 45.12 [30], dentistry students from Turkey with a mean score of 49.96 for females and 50.26 for males [31] or Spanish nursing students with scores exceeding a mean of 50 points [32]. Students of non-health related curriculums also had elevated STAI results, whereas Turkish female students had mean scores of 45.28 and males of 41.29 points [33]; Italian students had a mean score of 49.8 points [34]. Lower scores were revealed by Chinese students with a mean value of 39.5 points [29]. This proves the replicability of results and overall tendency of increased levels of anxiety among different populations studied by different researchers. Future studies need to verify if the norm of anxiety needs to be shifted towards higher values if this phenomenon remains as it is, or anxiety level of students will decrease as the pandemic ends.

The research conducted and the findings obtained indicate that there are relations between the variables of confirmed significance that confirm the fact that the type of university while also the feeling of social support and paying particular attention to nutrition during the pandemic (treated as a sign of pro-health behavior) may in varying degrees be associated with the level of anxiety felt.

On the basis of the data acquired, it is possible to say that the participation of the predictor, which is the worsening of the level of nutrition during the course of the pandemic (among men) to the formation of the level of anxiety is of a negative nature (this is indicated by the value of B). It is safe to state that the participation of this predictor, in which in the model of regression, other factors were also stipulated (such as paying particular attention to one’s diet, namely, the way of nutrition during severe anxiety felt during the pandemic and social support of close persons) was of a negative nature (Table 3). The interpretation was based on the indicator that the value over zero informs us that the growth of the value of the predictor (endogenous variable) is accompanied by the growth of value of the dependent variable in the case of the value below zero for the reverse. As presented in above results from other studies, gender differences of STAI items justify this type of division. Moreover, presented regression models with division for genders returns different significant predictors, which suggest that there are different ways of handling stressful situations not only in terms of current emotions and situations (state-anxiety) but also imprinted characteristics of individuals (trait-anxiety). These results only confirm the already known relation between anxiety level and gender, and the COVID-19 pandemic did not change these differences [35].

Different regression models were obtained with division by university. The best models were presented for Polish students. Lower number of curriculums made the target group more focused, which could explain the best fit among all models (R = 0.34 in comparison to around R = 0.20 of other models). S-anxiety model of Lviv State University of Physical Culture and t-anxiety model of National Academy of the National Guard of Ukraine had worse fit (R = 0.14 and 0.17), but p values of its predictors were close to significance level. More participants in those groups could lower the value and all its predictors could be significant.

Different specialization brings different education about health, physical activity [36,37,38] and dietary habits [39,40] of students. Fitness of regression model could be related to curriculum of students, where universities from Ukraine had students from specializations outside of health science studies. This could explain the worse fit of models, especially when dietary habits and poor diet while in distress were most commonly significant predictors in university comparison. Choice of career and studies is selected mostly by individual preference. If an individual is concerned about health-related topics, nutrition related predictors will be expressed in the chosen specialization. This could be considered as concurrent with personal psychological traits [41], which correspond with the trait-anxiety profile.

Support of a close person was significantly different according to both gender and university in the Kruskal–Wallis ANOVA analysis, but only in the case of males and National Academy of National Guard of Ukraine students was it a significant predictor for the regression model. This sense of support from partners [42], friends and coworkers [43] or family [44,45] played an important role in mental health and wellbeing during periods of lockdown and social distancing. This study did not specify what kind of close person the question was referring to, so different factors concerning social life could be taken into account. There are substantial reasons that let us think that better support of a close person is related to less anxiety. However, this predictor is accompanied by other factors in the model under consideration. One of the limitations of the current study is the lack of consideration in controlling for personality variables—in anxious situations, people with a high need for social affiliation require warm interpersonal relationships from those with whom they have close contact.

The last part of the study was composed of an exploratory analysis of predictor distribution impact on both state- and trait-anxiety variables. Students who had trait-anxiety levels above average values and state-anxiety below, varied the most in terms of predictor values.

As indicated by the value of the parameter “b”, the worsening of nutrition during the course of the pandemic fulfilled a relatively small and negative role among the pool of predictors of the model of regression. The worsening of nutrition during the course of the pandemic turned out to be a predictor that was negatively associated with the level of anxiety. It is possible to note that the worsening of nutrition during the course of the pandemic did not match the growth in the level of anxiety well, but none of the predictors formulated in the equation of regressions (the model of regression) operated independently.

On the graph with highest dispersion of scores, there were higher levels of trait-anxiety. Despite a strong association between social environment and stress and mental health [46], biological factors expressed in the personal traits of individuals played a crucial role in the anxiety level during stressful situations. Traits are related not only to genes and in-born capabilities of individuals, but also to the environment in which a person develops, where genetic and epigenetic traits for mental health are expressed upon social environment composed of relatives, close people, diet and education [47,48].

As this study revealed, some individuals may experience increased anxiety during pandemics. This state affects their attitude towards health-related behaviors, such as dietary habits, and makes it worse. These findings indicate that there may be a necessity for an interdisciplinary approach to mental health issues by psychologists, psychiatrists and dietitians. As the second aspect of this study found, specialists should work not only with the individual, but also with the person’s close social circle, as their support could play a role in prophylaxis or therapy.

## 5. Conclusions

This study demonstrated that both state- and trait-anxiety levels of university students were elevated in comparison to referential values, suggesting potential proneness to mental health problems due to the COVID-19 pandemic. Both forms of anxiety are important for everyday functioning, but anxiety levels are significantly different among genders of students and their universities. This study demonstrates that anxiety levels are multifaceted and more complex than a simple relation to dietary habits or sense of emotional support in the close social environment. We found that overall, the reciprocal associations between anxiety level and dietary habits, understood as worse nutrition during the pandemic than prior to the occurrence of COVID 19, were stronger for women students than for men. These results suggest the need to adopt an integrated approach toward university students and can be of importance for developing preventative health interventions. Mental health professionals should take into consideration working alongside with dietitians and a close social circle of individuals for better health anxiety management or treatment.

## 6. Limitations

Despite a relatively large sample, this study’s findings are limited. Use of more predictors, such as socioeconomic status or running additional personality tests, would bring more insight to both of the anxiety levels tested in this study. This study has exploratory value as a knowledge extension. Lack of psychological resilience testing is also limiting these findings. These findings are referring only to the general anxiety level of the studied population. We hope that on the basis of the concepts of health psychology, we have conducted research which reaches the new data in order to answer the question about the relation between the level of anxiety, social support felt and paying particular attention to nutrition during the pandemic.

## Figures and Tables

**Figure 1 ijerph-18-08785-f001:**
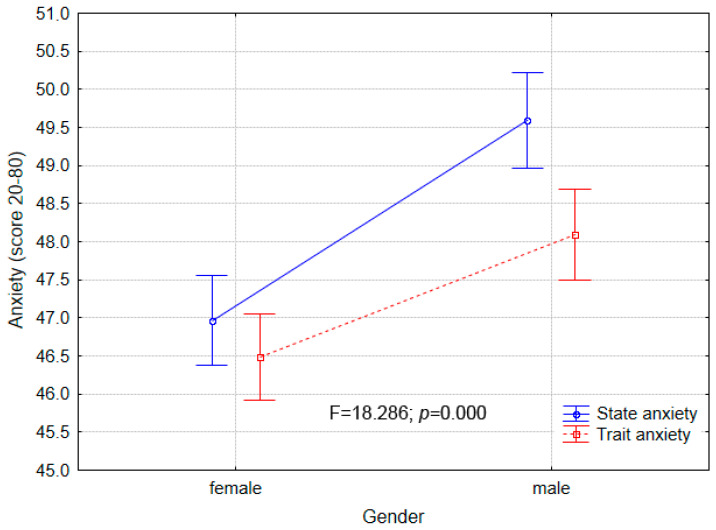
STAI items (state-anxiety and trait-anxiety) levels by gender of tested students.

**Figure 2 ijerph-18-08785-f002:**
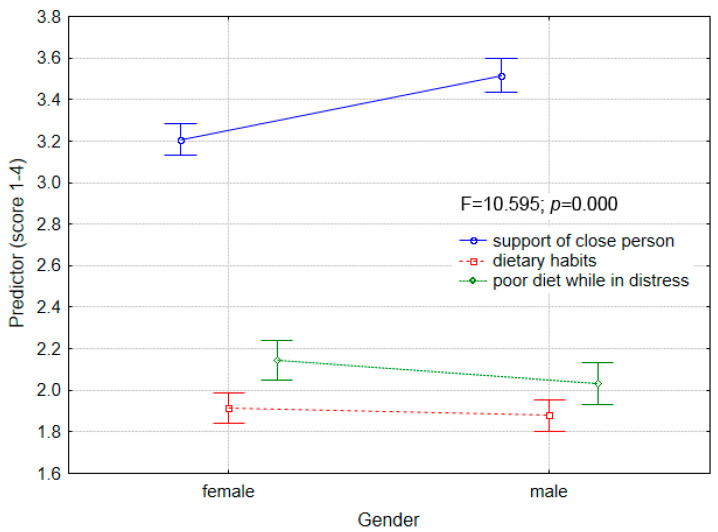
Descriptive statistic results of supplementary questions by gender.

**Figure 3 ijerph-18-08785-f003:**
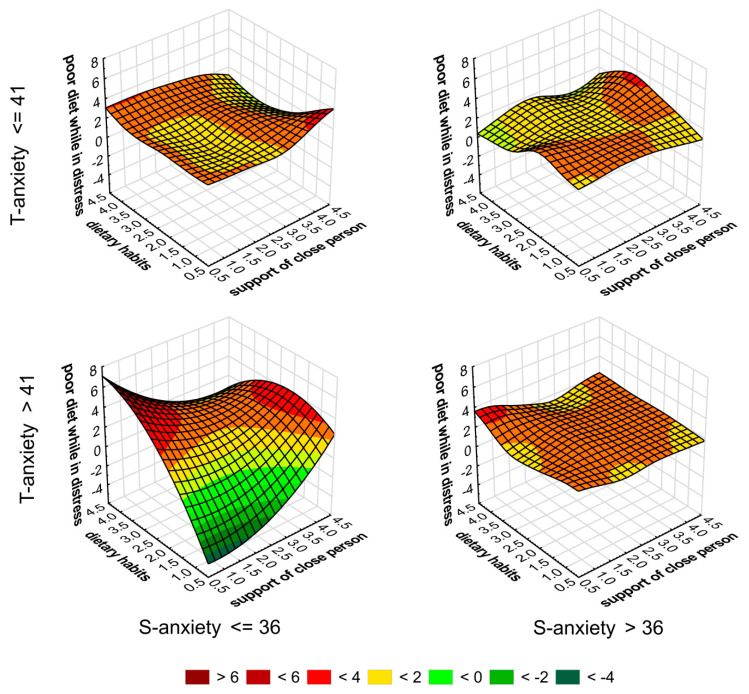
Categorized surface graph presentation of supplementary questions results divided into two variables of STAI items, with division into below and above average zones.

**Table 1 ijerph-18-08785-t001:** Characteristics of the participants by gender, university and specialization (*N* = 1012).

Gender	University	Specialization	*N*	Mean Age
Female (*N* = 535)	National Academy of the National Guard of Ukraine (*N* = 36)	Military administration	22	21.59
		Command and staff	14	22.07
	Lviv State University of Physical Culture (*N* = 197)	Physical education	69	19.42
		Physiotherapy	93	19.73
		Coach (sports)	35	20.00
	Jan Dlugosz University in Czestochowa (*N* = 302)	Physiotherapy	199	20.31
		Physical education	103	20.35
Male (*N* = 477)	National Academy of the National Guard of Ukraine (*N* = 258)	Law	68	21.35
		Military administration	48	21.18
		State security	142	22.12
	Lviv State University of Physical Culture (*N* = 144)	Physical education	41	20.19
		Physiotherapy	44	19.77
		coach (sports)	59	20.15
	Jan Dlugosz University in Czestochowa (*N* = 75)	Physiotherapy	43	20.79
		Physical education	32	20.96

**Table 2 ijerph-18-08785-t002:** Regression models grouped by gender of participants (*p* < 0.05).

Gender	Anxiety	Predictor	B	S	T(531)	*p*
Female (*N* = 535)	State-anxiety R = 0.20	support of close person	0.096	0.349	0.275	0.783
		dietary habits	0.190	0.405	0.469	0.639
		poor diet while in distress	**1.535**	**0.316**	**4.852**	**0.000**
	Trait-anxiety R = 0.13	support of close person	−0.193	0.324	−0.596	0.551
		dietary habits	0.457	0.375	1.218	0.224
		poor diet while in distress	**0.800**	**0.293**	**2.730**	**0.007**
Male (*N* = 477)	State-anxiety R = 0.20	support of close person	**0.927**	**0.298**	**3.115**	**0.002**
		dietary habits	0.420	0.249	1.685	0.093
		poor diet while in distress	−0.440	0.199	−2.218	0.027
	Trait-anxiety R = 0.20	support of close person	0.121	0.326	0.371	0.711
		dietary habits	**0.837**	**0.273**	**3.065**	**0.002**
		poor diet while in distress	**0.666**	**0.217**	**3.062**	**0.002**

B—unstandardized regression coefficient, indicates the average change in the dependent variable associated with a 1-unit change in the dependent variable, statistically controlling for the other independent variables; S—standard error of the regression; bold—statistically significant.

**Table 3 ijerph-18-08785-t003:** Regression models grouped by university of participants (*p* < 0.05).

University	Anxiety	Predictor	B	S	T(531)	*p*
National Academy of the National Guard of Ukraine (*N* = 294)	State-anxiety R = 0.27	support of close person	**1.737**	**0.389**	**4.467**	**0.000**
		dietary habits	0.100	0.285	0.351	0.726
		poor diet while in distress	−0.300	0.255	−1.180	0.239
	Trait-anxiety R = 0.17	support of close person	0.130	0.421	0.309	0.758
		dietary habits	**0.683**	**0.308**	**2.219**	**0.027**
		poor diet while in distress	0.539	0.275	1.958	0.051
Lviv State University of Physical Culture (*N* = 341)	State-anxiety R =0.14	support of close person	−0.211	0.353	−0.599	0.550
		dietary habits	0.733	0.396	1.850	0.065
		poor diet while in distress	−0.615	0.319	−1.930	0.054
	Trait-anxiety R = 0.23	support of close person	−0.431	0.306	−1.408	0.160
		dietary habits	**1.329**	**0.344**	**3.860**	**0.000**
		poor diet while in distress	0.409	0.277	1.478	0.140
Jan Dlugosz University in Czestochowa (*N* = 377)	State-anxiety R = 0.34	support of close person	**0.913**	**0.445**	**2.051**	**0.041**
		dietary habits	−0.767	0.513	−1.496	0.135
		poor diet while in distress	**2.383**	**0.359**	**6.638**	**0.000**
	Trait-anxiety R = 0.21	support of close person	0.582	0.436	1.335	0.183
		dietary habits	−0.816	0.502	−1.624	0.105
		poor diet while in distress	**1.334**	**0.352**	**3.793**	**0.000**

B—unstandardized regression coefficient, indicates the average change in the dependent variable associated with a 1-unit change in the dependent variable, statistically controlling for the other independent variables; S—standard error of the regression; bold—statistically significant.

**Table 4 ijerph-18-08785-t004:** Kruskal–Wallis ANOVA by gender and university differences of STAI items and supplementary questions (*p* < 0.05).

Variable	Category	*N*	Mean Rank	*p*
Support of close person				
Gender	Female	535	474.7019	*p* = 0.0000
	Male	477	542.1646	
University	National Academy of the National Guard of Ukraine	294	597.2347	*p* = 0.0000
	Lviv State University of Physical Culture	341	466.0323	
	Jan Dlugosz University in Czestochowa	377	472.3448	
Dietary habits				
Gender	Female	535	509.9	*p* = 0.6700
	Male	477	502.6	
University	National Academy of the National Guard of Ukraine	294	528.7	*p* = 0.0001
	Lviv State University of Physical Culture	341	539.6	
	Jan Dlugosz University in Czestochowa	377	459.2	
Poor diet while in distress				
Gender	Female	535	518.2	*p* = 0.1568
	Male	477	493.4	
University	National Academy of the National Guard of Ukraine	294	467.5	*p* = 0.0002
	Lviv State University of Physical Culture	341	490	
	Jan Dlugosz University in Czestochowa	377	551.8	
S-anxiety				
Gender	Female	535	438.9	*p* = 0.0000
	Male	477	582.4	
University	National Academy of the National Guard of Ukraine	294	528.9	*p* = 0.0000
	Lviv State University of Physical Culture	341	572.3	
	Jan Dlugosz University in Czestochowa	377	429.5	
T-anxiety				
Gender	Female	535	488.1	*p* = 0.0333
	Male	477	527.2	
University	National Academy of the National Guard of Ukraine	294	468.1	*p* = 0.0000
	Lviv State University of Physical Culture	341	591.5	
	Jan Dlugosz University in Czestochowa	377	459.6	

**Table 5 ijerph-18-08785-t005:** Post-hoc pair comparison for three-group analysis of Kruskal–Wallis ANOVA by university.

	Pairwise Comparison *p*-Value		
	National Academy of the National Guard of Ukraine	Lviv State University of Physical Culture	Jan Dlugosz University in Czestochowa
Support of close person			
National Academy of the National Guard of Ukraine		0.000	0.000
Lviv State University of Physical Culture	0.000		1.000
Jan Dlugosz University in Czestochowa	0.000	1.000	
Dietary habits			
National Academy of the National Guard of Ukraine		1.000	0.007
Lviv State University of Physical Culture	1.000		0.001
Jan Dlugosz University in Czestochowa	0.007	0.001	
Poor diet while in distress			
National Academy of the National Guard of Ukraine		1.000	0.001
Lviv State University of Physical Culture	1.000		0.014
Jan Dlugosz University in Czestochowa	0.001	0.014	
S-anxiety			
National Academy of the National Guard of Ukraine		0.187	0.000
Lviv State University of Physical Culture	0.187		0.000
Jan Dlugosz University in Czestochowa	0.000	0.000	
T-anxiety			
National Academy of the National Guard of Ukraine		0.0000	1.0000
Lviv State University of Physical Culture	0.0000		0.0000
Jan Dlugosz University in Czestochowa	1.0000	0.0000	

## Data Availability

The data presented in this study are available on request from the corresponding author.

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
