# Peer review of "Social Support and Dietary Habits as Anxiety Level Predictors of Students during the COVID-19 Pandemic"

_ijerph, 2021, doi:10.3390/ijerph18168785_

Round 1

Reviewer 1 Report

This study examines the relationship between dietary habits, social support and anxiety in a sample of Polish and Ukranian university students. The manuscript would benefit from thorough proofreading for grammar and flow by a native English speaker. For example, there are numerous sentences (e.g., “...expand knowledge on the subject of the relations that shape the levels of anxiety… in terms of the relations towards the sense of…”; p1, line 12-15) that are lengthy and difficult to follow. There are also various grammatical errors and awkward phrasing throughout the manuscript that can be improved on.

Introduction

The introduction can be structured more clearly. Currently, the introduction is difficult to follow as the current structure seems to alternate between talking about dietary habits and social support.

The authors can afford to expand on why eating behaviors and social support are important and examined in the current study. Although the authors provide some justification as to why these factors are examined (e.g., “It is crucial to maintain proper eating habits and physical activity, as well as positive social relations to maintain high quality of life” (p.1, line 34), these justifications are very brief.

Additionally, how these factors may be associated with improved well-being is unclear. While the authors state that “Role of social support during different stages of life was confirmed by a research on molecular level, where sense of support and social-economic status affects telomers length” (p.2, line 53), it is not clear what this implies.  

The authors mention that they aim to “verify how biological variables like a declared gender or university of students plays significant role in the anxiety level…” (p.2, line 64). However, prior to this line, the authors did not mention why gender and education level are examined in the current study.

Methods

Based on Table 1, N = 1,011 and not 1,012 as written in p.2, line 72.

The scale anchors are not mentioned. It is unclear what higher scores on the STAI and supplemental questions correspond to. Additionally, for the supplemental questions, it is unclear whether state or trait social support and dietary habits were assessed. As such, it is difficult to interpret the results presented later on.

The current study measures “positive dietary habits understood as healthy eating behaviors” and “bad diet understood as unhealthy eating behavior in distress and anxiousness” (p.3, line 102). It seems that the current study examines participants’ knowledge on dietary behaviors rather than participants’ dietary behaviours. However, the introduction and results sections appears to suggest that the current study examines anxiety and dietary behaviors.

The authors can afford to provide clearer description of the statistical analysis performed. For example, what software was used in the current analyses. Additionally, it is unclear what the authors mean by “Regression models were obtained in order to verify the shape of relations between studied variables” (p.4, line 113).

Results

While the authors provide a comprehensive breakdown of the between-group differences of the students from the different universities, it is unclear what significant implications these results have. Although students from different universities may have significantly different levels of perceived social support, these results do not seem to be generalizable to the wider, general population. However, it may be beneficial to divide the sample based on specialization rather than university, given that the authors mention that “Different specialization brings different education about health, physical activity [32-34] and dietary habits [35,36] of students” (p.11, line 260).

The authors should also provide the Standard Deviation of the key variables measures (besides the mean values).

One of the biggest limitation of the current study is the lack of consideration in controlling for potential confounds such as demographics (e.g., socioeconomic status) and personality variables. A separate analysis with a more comprehensive adjustment on potential covariates should be conducted.

Discussion

The argument that “Mean values were around 234 47-50 points, which exceeds average value for this age by almost 10 points” (p.11, line 234) can be strengthened by examining whether the difference is statistically significant by running a one-sample t-test.

The authors’ conclusion that “positive impact of social relation and proper eating habits 286 lowers the anxiety levels and allows students to better handle stressful situations.” (p.12, line 285) is premature, given that the current study is cross-sectional and cannot establish causality. It is possible that lower anxiety levels allow individuals to form better social relations and develop better eating habits.

Author Response

Thank you very much for your kindness and time. We appreciate that you gave us opportunity to correct our study. We tried our best to improve our work. Here we present detailed list of corrections and responses

Reviewer 2 Report

This is a good work with an excellent introduction and adequate methodology.

The title says "Social support and dietary habits as anxiety level predictors of students during COVID-19 pandemic ". However no information is collected about COVID-19 (fears of COVID-19, contact with COVID-19, confined time?, infection with COVID-19...). Only one sentence in discussion appears about COVID 19, and it says that seems "COVID-19 pandemic brings no difference".  Results are very similar with other studies, which were made also during the pandemic situation but, What about results of anxiety en student before pandemic situation? Only comparing with this studies you can affirm if the pandemic brings no difference. 

Using STAI it is a very good election, however measure whit only 1 item the supplementary variables seems very poor. What concrete answer where made? How do you know that are really good questions for measuring these aspects. More information is need it in order to be sure that the interpretation of the results are adequate.  No information about resilience was collected, it would have been very useful to determine if higher resilience determine lower anxiety levels. 

At what moment was made the data collection? Was the same pandemic situation in Ukraine and Poland? 

In table 2, line Male, poor diet while in stress, appears a p of .027. In the title says that p< .05, so this value means that it is statistically significant. Moreover, the beta value is negative. However is not in bold and no explanation is made about it. 

In table 3, in Lviv State University appears poor diet while in stress whit a p value of .054, it is true that is not significant, but is very near to, so I consider that should be mention. 

These both results are not mentioned in the discussion section. 

Looking tables 2 and 3 I feel confusion about the results. Even in the discussion authors say that a better support of close person is related with less anxiety, in table 2 and 3 when this variable is significant is in a positive way, this means that higher punctuation in support of close person, higher state anxiety.  As I mentioned before, in males a negative value in b of "poor diet while in distress", so higher values in this variables are associated to less state anxiety. These results are contradictory and needs a good explanation about their mean. 

In table 4, when 3 groups are compared, and significant results are obtained, it is necessary to calculated a post-hoc analysis in order to establish where the difference is or are. How do you calculate the "means rank"? You need to explain it. 

You do not present limitation section, so should be introduced. 

Author Response

We would like to thank you for highlighting issues that’s requires corrections and clarification. Following them, we have made changes referring to each remark received from the Reviewer, in the sequence of their appearance. We tried our best to answer to all your concerns and provide additional clarification in the text. 

Round 2

Reviewer 1 Report

The authors have adequately addressed my concerns. Congratulations!

Author Response

Dear Reviewer,
With warm the gratitude,  we would like to thank you very much for positive opinion about our work. We are delighted for opinion. We are pleased to see that we made satisfying changes and you are willing to accept our paper for further publishing process.

With all the gratitude, 
 Yours sincerely